# A Renewed Outbreak of the COVID−19 Pandemic: A Longitudinal Study of Distress, Resilience, and Subjective Well-Being

**DOI:** 10.3390/ijerph17217743

**Published:** 2020-10-23

**Authors:** Shaul Kimhi, Yohanan Eshel, Hadas Marciano, Bruria Adini

**Affiliations:** 1Stress and Resilience Research Center, Tel-Hai College, Northern Galilee 1220800, Israel; shaulkim@telhai.ac.il (S.K.); yeshel@psy.haifa.ac.il (Y.E.); hmarcia1@univ.haifa.ac.il (H.M.); 2Department of Psychology, University of Haifa, Haifa 3498838, Israel; 3Ergonomics and Human Factors Unit, University of Haifa, Haifa 3498838, Israel; 4Department of Emergency Management and Disaster Management School of Public Health, Sackler Faculty of Medicine, Tel Aviv University, Tel Aviv 6139001, Israel

**Keywords:** COVID-19, sense of danger, distress symptoms, perceived threats, individual, community and national resilience, well-being, hope and morale

## Abstract

Considering the potential impact of COVID-19 on the civil society, a longitudinal study was conducted to identify levels of distress, resilience, and the subjective well-being of the population. The study is based on two repeated measurements conducted at the end of the pandemic’s “first wave” and the beginning of the “second wave” on a sample (*n* = 906) of Jewish Israeli respondents, who completed an online questionnaire distributed by an Internet panel company. Three groups of indicators were assessed: signs of distress (sense of danger, distress symptoms, and perceived threats), resilience (individual, community, and national), and subjective well-being (well-being, hope, and morale). Results indicated the following: (a) a significant increase in distress indicators, with effect sizes of sense of danger, distress symptoms, and perceived threats (Cohen’s d 0.614, 0.120, and 0.248, respectively); (b) a significant decrease in resilience indicators, with effect sizes of individual, community, and national resilience (Cohen’s *d* 0.153, 0.428, and 0.793, respectively); and (c) a significant decrease in subjective well-being indicators with effect sizes of well-being, hope, and morale (Cohen’s *d* 0.116, 0.336, and 0.199, respectively). To conclude, COVID-19 had a severe, large-scale impact on the civil society, leading to multidimensional damage and a marked decrease in the individual, community, and national resilience of the population.

## 1. Introduction

The COVID-19 pandemic, which erupted in China in 2019, is an infectious disease caused by a newly discovered strain of coronavirus. This epidemic has spread rapidly worldwide. As of July 2020, an increasing prevalence of morbidity and mortality has been observed in several countries, while other countries present a consistent decline in the number of patients and deaths [1,2]. This pandemic has severely disrupted the proper functioning of the global community, leading to the closure of schools and academic institutions, partial or complete lockdowns, reduced public transportation and aviation, unemployment and economic hardships, decline of global stock markets, and panic shopping due to widespread concerns about supply shortages [3]. The restrictive measures that have been implemented by many governments to protect public health have substantially impacted the levels of distress, resilience, and subjective well-being of civil societies. These three concepts, which substantially influence on the capacity of any society to cope with adversities such as pandemics, will be explained.

**Distress.** Diverse adversities often give rise to post-traumatic symptoms, which may take the form of delayed emotional and behavioral problems [4], or depression, anxiety, grief, and post-traumatic stress disorder (PTSD) [5].

Research indicates that social threats—such as reducing the capacity for social support—can be caused by crises, negatively affecting people’s health and well-being [6]. Several studies following the SARS epidemic (which broke out in 2003) have indicated a continuing increase in distress symptoms of post-traumatic stress disorder and altruistic emotions [7]. Moreover, recent studies have shown that COVID-19 causes an increase in distress [8].

**Resilience.** A considerable number of definitions of the concept of resilience can be found in the professional literature [9]. Masten (2018) defines resilience as “the potential of the manifested capacity of a dynamic system to adapt successfully to disturbances that threaten the function, survival, or development of the system” (p. 187). The American Psychological Association defines resilience as a process of bouncing back from difficult experiences and adapting well in the face of adversity, trauma, tragedy, threats, or significant sources of stress [10]. Overall, researchers seem to agree on two main issues: First, the concept of resilience has often been used in discussing people’s ability to withstand stress and adversity [11]. Second, resilience is a complex multifaceted concept whose measurement arouses debate among researchers [12]. Experts have agreed that resilience is a complex construct that is defined differently in the context of individuals, families, organizations, societies, and cultures [13]. Three types of resilience have been extensively studied: individual, community, and national resilience.

Cacioppo, Reis, and Zautra [14] define *individual resilience* as “the capacity to foster, engage in, and sustain positive relationships and to endure and recover from life stressors and social isolation” (p. 44). Bonanno, Romero, and Klein [12] report that individual resilience contributes significantly and negatively to the prediction of depression, anxiety, stress, and obsessive-compulsive symptoms. Furthermore, according to these researchers, *community resilience* expresses the interaction between individuals and their community and refers to the success of the community to in providing for the needs of its members and the extent to which individuals are helped by their community. *National resilience* is a broad concept addressing issues of social sustainability and strength in several diverse realms: trust in the integrity of the government, the parliament, and other national institutions, belief in social solidarity, and patriotism [15]. Examination of the relevant literature indicates that a rather small number of studies have empirically investigated national resilience and associated it with antecedent [16,17].

There is little information in the literature about the effects of a pandemic on the level of the different modes of resilience variables. Bonanno et al. [18] reported that those who have shown a greater level of resilience and recovered faster from a disease are those who have received a greater degree of social support. Other researchers report that adequate leadership, which encourages emotional and cognitive organization (such as a clear explanation of the problem and how to deal with it), facilitated an increase of the population’s resilience during an epidemic [19].

**Subjective well-being.** According to Naci and Ioannidis [20], wellness refers to three interconnected dimensions of physical, mental, and social well-being that extend beyond the traditional definition of health. The literature on subjective well-being (SWB) focuses mostly on happiness, life satisfaction, and positive affect. However, there is no agreement among researchers concerning the components that constitute SWB.

A recent literature survey indicates that there are 45 different ways and at least 63 different constructs that have been employed in studying SWB [21]. Studies have indicated that COVID-19 impairs the standard of living due to various limitations resulting from attempts to combat the epidemic [22]. However, Sibley et al. [23], does not report any decline of well-being due to the COVID-19 pandemic lockdown in New Zealand.

The aim of the study was to investigate the impact of the COVID-19 pandemic on the levels of distress, resilience, and subjective well-being of the population over time. Thus, we conducted a longitudinal study (the same sample answers the same questionnaire at two time-frames, among the Israeli public to examine three groups of indicators: distress, resilience, subjective well-being. To the best of our knowledge, no study has yet examined the effects of the coronavirus on the level of these three groups of indicators in a sample of a country’s population. The first measurement was made at a time of reduced restrictions on the population, such as the lifting of the lockdown, and the overall perception was that the crisis was over, while the second measurement was conducted during the renewed outbreak of the crisis and severe aggravation of the economic situation. Therefore, we hypothesize the following:Resilience indicators will significantly correlate with SWB indicators, and both will significantly and negatively correlate with distress indicators, across the two measurements. These correlations will show medium effect size compared to pre COVID-19 levels.Resilience and SWB indicators will significantly decrease between T1 and T2. These decreases will show medium effect size compared to pre COVID-19 levels.Distress indicators will significantly increase between T1 and T2. These increases will show medium effect size compared to pre COVID-19 levels.

## 2. Materials and Method

### 2.1. Study Design

To investigate the impact of the COVID-19 pandemic on the levels of distress, resilience, and subjective well-being of the Israeli population, a longitudinal study was designed based on two repeated measurements. The study was performed on a large sample of respondents who answered the same questionnaire at two repeated measurements (paired sample). The first measurement (T1) was carried out in early May 2020 (4–7 May), when the first wave of the pandemic seemed to recede, and the full lockdown and other limitations on the population were lifted. The second measurement (T2) was conducted in mid-July 2020 (12–15 July) with the re-emergence of the pandemic in Israel (the “second wave”), which led to the re-imposition of restrictions on the population.

Three groups of indicators were measured in both measurements: distress (sense of danger, distress symptoms, and perceived threats), resilience (individual, community, and national resilience), and subjective well-being (well-being, hope, and morale), assuming that a pandemic of this magnitude would affect all these three groups of indicators.

To better understand the trends of the population’s resilience, distress, and subjective well-being over time, the current two repeated measurements were compared with previous research results, which were based on a representative sample of the Israeli population. These findings show the resilience and distress levels of the Israeli population as found in 2018, during a relatively quiet period in terms of security risks, thus representing a baseline measurement [24].

The data were collected by an internet panel company that consists of over 65,000 panelists, representing all demographic sectors and geographic locations (https://sekernet.co.il/). A stratified sampling method was used, which is aligned with the data published by the Israeli Central Bureau of Statistics, to appropriately include the varied groups of the Israeli population concerning gender, age, and geographic dispersal. The utilization of internet panels has been increasing rapidly, and its validity has been widely discussed [25].

### 2.2. Participants

The participants in both measurements were a paired sample of Jewish Israeli respondents who answered an online questionnaire distributed by an Internet panel company. The first measurement (answered by *n* = 1100) was conducted during 4–7 May, with the release from the full lockdown imposed on Israeli residents. During that week, all the individuals tested for the virus were found to be negative for COVID-19 (Surveillance of COVID-10 in Israel, 2020) [26]. The second measurement (answered by 82% of the original sample; *n* = 906) was conducted between 12 and 15 July 2020. In this period, the rising numbers of confirmed COVID-19 patients (2.2% of the individuals tested for the virus were found to be positive for COVID-19), increasing the probability of the re-issuing of restrictive measures to combat the pandemic, including lockdowns. The demographic and psychological characteristics of the sample population are detailed in Table 1.

### 2.3. Study Tools

All the questionnaires employed in the present study have been used by us in previous studies with one difference: Wherever the original item in the questionnaire referred to a security threat and/or a security situation, the wording of the item was modified to suit the coronavirus crisis. The following scales are included in this study.

**Sense of danger****.** This scale pertains to the level of the individual, social, and national sense of danger [27]. Four additional items that focus on the COVID-19 have been added to the original 7 items: e.g., “To what extent are you afraid that you will have difficulty finding work after the corona crisis?” or “To what extent are you afraid that you will not have anyone to help you financially?” These 11 items were rated by a scale ranging from 1 = not at all to 5 = very much. The scale’s Cronbach’s alpha reliability in the current study was good: α = 0.86 (T1) and α = 0.88 (T2).

**Distress symptoms**. Two subscales of the Brief Symptom Inventory (BSI) scale were employed in the present study: anxiety (3 items) and depression (5 items) [28]. The item about suicidal thoughts was removed from this scale for ethical reasons. Respondents were asked to report the extent to which they are currently suffering from any of the problems presented. The responses range from 1 = not at all to 5 = to a very large extent. The internal reliabilities of these scales in the present study were high: α = 0.91 (T1) and α = 0.92 (T2).

**Perceived threats**. A threat is a potential danger of harm to an individual as perceived by the individual [29]. A threat can be defined as potential damage. The threat can be related to different areas, e.g., physical, social, psychological, economic, and more. In the present study, we asked the respondents to rate four different threats: economic, health, security, and a threat arising from the political situation in Israel. The political threat is important because Israel has faced three rounds of election with no clear-cut balance between parties during the year that preceded the coronavirus pandemic eruption. The third election was carried out on 2 March 2020, approximately one week before the World Health Organization (WHO) declared that COVID-19 could be characterized as a pandemic (11 March 2020, see https://www.who.int/news-room/detail/27–04-2020-who-timeline---covid-19). Thus, the period of the coronavirus pandemic in Israel was also characterized by political instability. The answers to this question constitute a 5-point scale, ranging from 1 = not threatening at all to 5 = threatening to a very large extent. We have used the *sum* of all 4 threats as an index for threat perception.

**Individual resilience**. The short version of this questionnaire [30,31] includes 10 items about a sense of personal resilience in the face of difficulties. Examples of questions are as follows: “I am able to adapt when changes occur”; “I am not easily discouraged by failures”. Responses to the questionnaire items are ranked using a 5-point scale ranging from 0 = not true at all to 4= true nearly all the time. In the present study, the internal scale reliability of the scale was high in both measurements: α = 0.89 (T1) and α = 0.90 (T2).

**Community resilience**. This resilience scale includes 10 items that relate to the subjects’ identification with their community and their confidence in their ability to cope with the difficulties they will face [32]. Responses to the questionnaire items represent a 5-point scale, ranging from 1 = do not agree at all to 5 = agree to a very large extent. Examples of items are as follows: “The municipal authority in my locality is functioning properly in the corona crisis”, “I can trust people in my locality to come to my aid in case of a crisis, including the corona crisis”. The current study internal scale reliabilities were high in both measurements (α = 0.93).

**National resilience** (NR). The original national scale [16] includes 13 items, whereas the scale in the present study includes 16 items. The three additional items pertain specifically to the COVID-19 crisis. Examples of the original scale items are as follows: “In a national crisis, the Israeli society will stand behind the decisions of the government and its leader,” and “Israel is my home and I do not intend to leave it”. An example of a new item is, “I have full confidence in the ability of the Israeli healthcare system to take care of the population during the coronavirus crisis.” The response scale for the national resilience items ranges from 1 = do not agree at all to 6 = strongly agree. The internal reliability of the scale was high in both measurements (α = 0.91).

**Well-being**. This scale consists of nine items concerning individuals’ perception of their lives in the present regarding various contexts, such as work, family life, health, free time, and others [33]. This scale is based on the recovery scale we have used in previous studies. Responses to these items range from 1 = very bad to 6 = very good. This measurement scale has been validated in previous studies, and its reliability in the present study was found to be good in both measurements (α = 0.87).

**Level of hope.** This tool, which has been constructed specifically for the present study, is based on an earlier study [34,35] that was designed to measure the level of hope for peace between Israel, the Arab nations, and the Palestinians. Its two dimensions are personal and collective hope. The current scale of hope, in the context of coronavirus, includes five items. Two of them refer to the personal level (e.g., “I hope that I will emerge strengthened from the coronavirus crisis”) and three items refer to the collective level (e.g., “I hope that Israeli society will emerge strengthened from the coronavirus crisis”). The internal reliability of the scale in the present study was found to be high in both measurements (α = 0.92).

**Morale.** The level of personal morale was examined by a single item: “How would you define your morale these days?” The response scale ranges from 1 = not good at all to 5 = very good.

**Demographic characteristics**. Respondents reported eight demographic variables: age (18–30, 31–40, 41–60. 60 +), gender (1 = male, 2 = female), level of religiosity (1= non-religious to 4 = very religious), family income relative to the average income in Israel (1 = much lower than the national average to 5 = much higher than the national average), political attitudes (1 = extreme left to 5 = extreme right), level of education (1 = elementary to 5 = graduate degree and higher), familial status (single, married, divorced, couple), number of children (no children to 4 children or more). Table 1 presents the distribution of these attributes among the present sample.

### 2.4. Data Analysis

We used 4 statistical calculations to examine our hypotheses: (a) We calculated Pearson correlations between the study variables in each of the two repeated measurements. (B) We analyzed the differences in the psychological variables between the two measurements, using the General Linear Model for two repeated measures, further calculating the effect sizes. (c) To examine the structure of the national resilience and the differences between the two measurements, we performed a factor analysis on the national resilience items and examined the difference in each factor in the two measurements using a General Linear Model. (D) Finally, to measure the difference in the four perceived threats between the two measurements, we used the General Linear Model analysis.

## 3. Results

To examine our first hypothesis, we calculated the correlation matrix among the nine investigated psychological variables (Table 2). Table 2 indicates the following. (a) All the correlations are significant (*p* < 0.001) in both measurements (T1 and T2). (b) The three distress indicators (sense of danger, distress symptoms, and overall threats) significantly and positively correlate with each other and significantly negatively correlate with the three resilience and three subjective well-being (SWB) indicators, across T1 and T2. (c) The three resilience and SWB indicators significantly and positively correlate with each other across T1 and T2. (d) The correlation between the variables in T1 and T2 is very similar. These results fully support our first hypothesis.

To examine our second and third hypotheses regarding the difference between the investigated variables in T1 and T2, general linear model analyses for two repeated measures have been calculated (Table 3). Results indicate the following: (a) There is a significant difference between T1 and T2 mean scores of all the nine examined variables. (b) The three resilience indicators decreased significantly between T1 and T2 (*p* < 0.001). (c) The three distress indicators increased significantly between T1 and T2 (*p* < 0.05). (d) The three SWB indicators decreased significantly between T1 and T2 (*p* < 0.001). (e) The largest decrease among the resilience indicators occurred in the level of national resilience (as can be seen by the largest effect size: *Cohen’s d*= 0.793). The largest increase among the distress indicators was found in the sense of danger (*Cohen’s d*= 0.614). The largest decrease among the quality of life indicators was related to hope assessments (*Cohen’s d* = 0.336). Table 3 also presents the differences between five variables that were originally collected among a different national sample at the end of 2018 (a relatively calm period security-wise, referred to as a “baseline” pre COVID-19 measurement), with their current corresponding variables at T2. Results indicate that compared with this baseline measurement (2018 national sample), the level of distress symptoms in T2 is higher, whereas the levels of resilience and well-being are lower. These results fully support our hypotheses.

To better understand the decrease in the levels of national resilience, we launched a factor analysis (principal component and Varimax rotation) on the NR scale of T2 and compared the mean of each item and factor with its corresponding mean item score in T1 (Table 4). Four factors emerged and were labeled as “trust in the state and its leader”; “trust in the Israeli society”; “patriotism”; and “trust in the public institutions of Israel”. The overall variance explained by these four factors is 69.39% of the total variance of the national resilience variable. All four factors decreased significantly from T1 to T2 (*p* < 0.001). The highest decrease was presented in factor 1: “trust in the state and its leader”.

An examination of the differences in the levels of perceived threats between T1 and T2 (Table 5) indicates a significant increase in each of the following threats: political, economic, health, and security (*p* < 0.001). Furthermore, the results indicate that the political threat was perceived as the highest risk, followed by economic, health, and security risks, in both periods. Nonetheless, a bigger effect size was noted in the difference between the two measurements concerning the health threat (Cohen’s d = 0.272).

## 4. Discussion

The present study examined indicators of distress, resilience, and subjective well-being, using two repeated measurements among a paired sample of respondents. We hypothesized that resilience and SWB would be positively associated with each other and negatively associated with distress indicators. We also hypothesized that the resilience and SWB would decrease, while distress indicators would increase, between the two measurements, due to the growing impact of the prolonged COVID-19 pandemic crisis. Overall, our results supported our three hypotheses and revealed that among the civil society, positive indicators such as resilience, morale, hope, and well-being weakened in T2 in relation to both T1 and to the pre COVID-19 pandemic. The negative indicators, such as sense of danger, distress symptoms, and perceived threats, rose in T2 in relation to both T1 and to the pre COVID-19 pandemic. The first measurement was conducted in early May 2020 at the end of a long lockdown imposed on Israeli residents, when the country seemed to have emerged from the crisis after the reduction in new cases following the “first wave” of the coronavirus outbreak. The second measurement was carried out two months later, in mid-July, at a time when the coronavirus pandemic emerged once more and was perceived as a threat to all Israelis. According to the findings of the current study, all indices that may contribute toward an effective coping with the crisis, without exception, significantly weakened during T2 compared with T1, and also compared to a relatively calm period in 2018 (pre COVID-19 pandemic).

The substantial decrease in the national resilience level between the two measurements, compared to the less drastic decrease presented for individual and community resiliencies, seems to represent a loss of trust in governmental bodies and the country’s leader. This finding is understandable, considering previous findings that highlighted that the adaptation of governmental institutions to the changing needs of the population is essential to national resilience [36,37]. The perception that the expected adaptation of governmental entities did not materialize sufficiently is strengthened by the large-scale demonstrations against the government that spread throughout Israel. Furthermore, a previous study indicated that the baseline level of patriotism was extremely high among Israelis [16]. However, an unexpected finding of the present study was a general decrease in patriotism between T1 and T2. The respondents appear to lose some of their faith in the country that does not appear to come to their aid in the current complex economic conditions, and as a result, they are less optimistic about the future of the country in general and their status specifically.

In addition to the marked decrease in national resilience, the results also indicate a significant decrease in individual and community resilience, albeit to a lesser extent. The decreased community resilience is somewhat surprising, as previous studies have presented a rise in the levels of community resilience during periods of elevated risks [38]. Furthermore, an earlier study that was conducted in 2018, following a military clash between the Israeli Defense Forces and the Palestinian Islamic Jihad terror organization in Gaza Strip, indicated that the average community resilience of the southern participants (residing closest to the high-risk area) was significantly higher compared to the national sample [38]. Another study that examined the Israeli population’s resilience during the peak of the COVID-19 pandemic, which included an overall lockdown versus the initial phase of lifting the lockdown, indicated no change regarding both individual and community resilience [39]. Based on these studies, it seems appropriate to claim that the decline of individual and community resilience in the current study is exceptional and reflects the above-mentioned lack of belief in governmental institutions, as well as in other authorities that are responsible for managing the economic, social, and educational ramifications of the COVID-19 pandemic.

The distress indicators that were examined in the current study (sense of danger, distress symptoms, and perceived threats) showed, as expected, a significant increase between the two measurements. In a previous study, which dealt with the impact of the coronavirus crisis compared to a pre-crisis baseline measurement [39], we found that a sense of danger is a more “sensitive” measure of the distress. Similarly, in the present study, we found that the gap between the two measurements was larger concerning feelings of danger compared to the distress symptoms and perceived threats. Further research is required to assess whether, when the coronavirus pandemic is contained, the decline of these three indicators will take place at similar or different rates. It is important to note that the sense of danger should be regarded as an expected response to threatening situations, as it serves as a warning mechanism targeted to alert people of possible harm that may be caused to individuals or their environment. Such a sense of danger may be accompanied by a parallel attempt to minimize stress responses and reinforce a sense of “control” to maintain proper functioning [40].

In the present study, the level of hope was found to be significantly and positively correlated with resilience, and negatively correlated with distress, and it was found to be significantly lower in T2 compared to T1. In our opinion, the level of hope emphasizes the importance of the psychological dimension in managing the COVID-19 crisis. Hope refers to the anticipation of a better future [41,42]. In an ongoing crisis that is characterized by great uncertainty, it may be difficult for people to anticipate an improved future. The positive correlation that was found between hope and the level of morale indicates further that people experience psychological difficulties that affect hope and despair.

All four levels of examined threats—economic, health, security, and political risks—increased from T1 to T2. The highest threat perceived by respondents was the political threat, followed by economic, health, and security threats. These results reflect the current fragile political situation that characterizes the Israeli society, which may lead to a fourth election within two years.

The increase in the economic threat expressed the fact that many self-employed people in Israel were severely impacted as a result of the coronavirus pandemic and have not as yet received the expected economic assistance from the state. A substantial number of people became unemployed or are on “unpaid vacation”, and although they received some financial compensation from the state, they still feel insecure or uncertain about their financial situation. Similar to findings from other countries [43], the economic threat remains the second highest of the four threats examined, even among individuals who were not directly hit by this pandemic. The fear of what the future may hold economically and otherwise is significant.

Regarding the security threat, there has also been an increase in its perception, even though apparently, and perhaps due to the coronavirus, the security situation in Israel is at present relatively improved. These results may perhaps be derived from the reports that have been broadcast in the past weeks in the media concerning mysterious attacks and explosions in Iran and Syria, which may be attributed by some to the actions of the State of Israel. These reports may raise concerns among people about future security developments to come.

As in any other research, this study is not exempt from limitations. The most notable limitation is the reliance on the internet panel sample on which this study is based. Despite the large sample and the widespread distribution of all demographic variables, it is not possible to guarantee that this sample represents the adult population in Israel. Nonetheless, as previously noted, internet panels have been frequently used in studying the varied phenomenon among the population, and their validity has been widely studied and confirmed.

Even though the current study was conducted in Israel, we believe that the findings may also apply to other countries dealing with the COVID-19 pandemic crisis, most especially as the civil societies in many countries were substantially impacted, including the healthcare, economic, and societal systems. This assumption is strengthened by our findings that the varied groups among the Israeli society, such as sectors with different economic levels, political attitudes, or religious beliefs, show similar trends. Furthermore, preliminary findings from a recent study that was conducted (based on the same tools) in Brazil and the Philippines present similar results regarding the relationships between resilience, SWB, and distress indices [44]. Additional studies are needed to support this assumption.

## 5. Limitations and Conclusions

Four notable limitations of this study should be mentioned. The first limitation is that the sample is based on a web sample and not on a random sample. The second limitation is the fact that this is a correlative study that does not allow inference to be derived. The third limitation of this study is the fact that the study was conducted in Israel; as no parallel longitudinal study was conducted thus far in other countries, it is difficult to generalize the findings, most especially considering varied biases and cultural diversities. The fourth limitation is that the health status of the respondents before COVID-19 was not studied and thus was not controlled for a possible mediating effect.

The main conclusion of the current study is that the coronavirus crisis has had a severe and large-scale impact on the civil society. It has caused multidimensional damage to the population (with health, economic, political, and social ramifications), which in turn has caused a marked decrease, with substantial effect sizes, in the national, community, and individual resilience of the population. The negative impact of the COVID-19 pandemic on levels of distress and resilience is more severe compared to crises that result from security risks. Based on the results of this study, it seems that Israeli society’s ability to deal with a prolonged crisis has weakened. It is recommended that such longitudinal studies be continued, both in Israel and in other societies, to identify trends in the capacities to deal with continued crises and better understand the factors that may empower or weaken the societal resilience of civil populations.

## Figures and Tables

**Table 1 ijerph-17-07743-t001:** Demographic characteristics of the respondents (*n* = 906).

Variable	Group	No. of Respondents	%	Average (S.D)
Age	18–30	211	23	44.08 (15.53)
	31–40	212	23	
	41–60	164	18	
	51–60	156	17	
	61–70	125	14	
	18–30	38	4	
Gender	Male	464	51	
	Female	442	49	
Level of religiosity	Secular	437	48	
	Traditional	269	30	
	Religious	117	13	
	Very religious (orthodox)	83	9	
Family income relative to average in Israel	1. Much lower	241	27	2.49 (1.20)
	2. Lower	234	26	
	3. About average	222	25	
	4. Above	164	18	
	5. Much above	45	5	
Political attitudes	1. Strong left	9	1	3.50 (0.85)
	2. Left	104	11	
	3. Center	304	34	
	4. Right	403	44	
	5. Strong right	86	9	
Education	1. Elementary	6	1	3.28 (1.01)
	2. High school	227	25	
	3. Above high school, no B.A	314	35	
	4. B.A.	228	25	
	5. M.A. and above	131	14	
Family status	1. Bachelor	210	23	
	2. Married	565	62	
	3. Divorce	75	8	
	4. Widow	9	1	
	5. living in partnership	47	5	
Number of children	1. No children	293	32	1.98 (1.90)
	2. One child	95	10	
	3. 2–3 children	367	41	
	4. 4–5 children	115	13	
	5. 6 and above	36	4	

**Table 2 ijerph-17-07743-t002:** Pearson correlations among distress, resilience, and subjective quality of life indicators (*n* = 906).

		Distress Indicators			Resilience Indicators			Subjective Well-Being Indicators		
Variables	Time	1	2	3	4	5	6	7	8	9
**1. Sense of Danger**	T1	--	0.529 ***	0.611 ***	−0.291 ***	−0.186 ***	−0.187 ***	−0.425 ***	−0.229 ***	−0.390 ***
	T2	--	0.540 ***	0.664 ***	−0.319 ***	−0.194 ***	−0.212 ***	−0.449 ***	−0.281 ***	−0.416 ***
**2. Distress symptoms**	T1		--	0.430 ***	−0.446 ***	−0.204 ***	−0.209 ***	−0.623 ***	−0.330 ***	−0.683 ***
	T2		--	0.494 ***	−0.450 ***	−0.240 ***	−0.252 ***	−0.615 ***	−0.357 ***	−0.678 ***
**3. Overall threats**	T1			--	−0.301 ***	−0.170 ***	−0.321 ***	−0.417 ***	−0.321 ***	−0.340 ***
	T2			--	−0.285 ***	−0.202 ***	−0.286 ***	−0.431 ***	−0.320 ***	−0.410 ***
**4. Individual resilience**	T1				--	0.283 ***	0.205 ***	0.495 ***	0.391 ***	0.499 ***
	T2				--	0.276 ***	0.211 ***	0.496 ***	0.390 ***	0.449 ***
**5. Community resilience**	T1					--	0.465 ***	0.310 ***	0.380 ***	0.286 ***
	T2					--	0.454 ***	0.362 ***	0.336 ***	0.288 ***
**6. National resilience**	T1						--	0.306 ***	0.440 ***	0.255 ***
	T2						--	0.309 ***	0.451 ***	0.336 ***
**7. Well-being**	T1							--	0.492 ***	0.687 ***
	T2							--	0.469 ***	0.644 ***
**8. hope**	T1								--	0.433 ***
	T2								--	0.460***
**9. Morale**	T1									--
	T2									--

*** *p* < 0.001.

**Table 3 ijerph-17-07743-t003:** General Linear Model—two repeated measures of distress symptoms, resilience, and subjective well-being—differences, and effect size (*n* = 906).

	Variable/Scale	T1	T2	Difference between T1 and T2	Effect Size	Compared 2018 Sample (*n* = 1100) with T2		Effect Size for T-Test
		M (SD)	M (SD)	*F* _(1,905)_	Cohen’s d	M	(S D)	Cohen’s d
**Distress**	Sense of danger (1–5)	2.62 (0.73)	2.94 (0.81)	342.49 *** **Increase**	0 0.614	2.57	0.74	
						t = 9.29 ***		0.476
	Distress symptoms (1–5)	2.27 (0.88)	2.36 (0.92)	14.25 * **Increase**	0.120	1.95	0.79	
						t = 10.73 **		0.478
	Overall sum of threats (1–5)	10.87 (3.23)	11.59 (3.28)	55.30 *** **Increase**	0.248	No data		
**Resilience**	Individual resilience (1–5)	2.48 (0.70)	2.40 (0.72)	23.21 *** **Decrease**	0..153	No data		

	Community resilience (1–5)	3.33 (0.88)	3.09 (0.87)	153.70 *** **decrease**	0.428	3.11	0.86	0.023
						t = 0.52 *n*.s.		
	National resilience (1–6)	3.84 (0.88)	3.34 (0.89)	562.48 *** **decrease**	0.793	3.95	0.92	
						t = 3.89 ***		0.673
**Subjective well-being**	Well-being (1–6)	4.17 (0.90)	4.09 (0.89)	11.69 *** **decrease**	0.116	4.68	0.82	
						t = 16.47 **		0.689
	Hope (1–5)	3.61 (0.91)	3.32 (0.99)	103.38 *** **decrease**	0.336	No data		
	Morale (1–5)	3.51 (0.93)	3.34 (0.92)	37.64 *** **decrease**	0.199	No data		

* *p* < 0.05, ** *p* < 0.01, *** *p* < 0.001, *n.s.* = not significant.

**Table 4 ijerph-17-07743-t004:** Factor analysis of national resilience (scale 1–6).

	Measure 1	Measure 2	Factor Loaded	Difference between T1 and T2	*F* _(1,905)_
					Effect Size
Factors and items and factor explained variance	M	M			
**Factor 1: Trust in the state and its leader, 25.11%**	**3.50**	**3.06**		0.44	**634.50 *****
1. I believe that my government will make the right decision during a time of crisis, including the current coronavirus crisis.	3.96	2.97	**0.796**	0.99	ηp^2^ = 0.41
2. During a national crisis, such as the current coronavirus crisis, society in my country will back up the government decisions and those of the prime minister.	3.77	2.80	**0.710**	0.97	
12. I have complete confidence in the ability of my government to take care of all aspects relevant to overcoming the current coronavirus crisis.	3.78	2.90	**0.777**	0.88	
11. I have full faith in the ability of my country’s health system to care for the population in the current coronavirus crisis.	4.05	3.36	**0.677**	0.69	
3. I have full confidence in the ability of the security forces of my country to protect our population including the current coronavirus crisis.	4.49	3.82	**0.502**	0.67	
14. Trust in the parliament (Knesset)	2.85	2.54	**0.557**	0.31	
**Factor 2: Trust in the Israeli society, 17.41%**	**3.45**	**3.29**		**0.16**	**99.77 *****
8. In my society, there is a high level of social solidarity (mutual assistance and concern for one another).	4.19	3.80	**0.713**	0.39	ηp^2^ = 0.10
7. Social relations between the different groups in my country are good.	3.17	2.84	**0.709**	0.33	
10. In my society, there is a reasonable level of social justice.	3.37	3.11	**0.684**	0.26	
9. The expression “man is a wolf to man” is ***not*** characteristic of my society.	3.59	3.42	**0.818**	0.17	
**Factor 3: Patriotism, 15.65%**	**4.79**	**4.33**		**0.46**	**251.76 *****
5. My society has coped well with past crises and will cope well with the current coronavirus crisis.	4.71	3.99	**0.699**	0.72	ηp^2^ = 0.22
6. I am optimistic about the future of my country.	4.51	3.98	**0.639**	0.53	
4. My country is my home, and I do not intend to leave it.	5.17	5.03	**0.832**	0.14	
**Factor 4: Trust in public institutions in Israel, 11.21%**	**4.79**	**4.33**		**0.31**	**132.19 *****
13. Trust in the police	3.45	2.85	**0.769**	0.60	ηp^2^ = 0.13
15. Trust in the education system	3.49	3.26	**0.498**	0.23	
16. Trust in the media	2.86	2.76	**0.836**	0.10	

*** *p* < 0.001.

**Table 5 ijerph-17-07743-t005:** General Linear Model: two repeated measures of four threats.

Type of Threat	T1		T2		*F* _(1,905)_	Repeated Measures Effect Size
	M	S.D	M	S.D		Cohen’s d
Political	3.11	1.30	3.31	1.28	22.30 ***	0.157
Economic	2.88	1.21	3.00	1.22	11.83 ***	0.177
Health	2.44	1.09	2.72	1.10	63.50 ***	0.272
Security	2.44	1.05	2.57	1.04	13.37 ***	0.124

*** *p* < 0.001.

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
