# Peer review of "A Renewed Outbreak of the COVID−19 Pandemic: A Longitudinal Study of Distress, Resilience, and Subjective Well-Being"

_ijerph, 2020, doi:10.3390/ijerph17217743_

Round 1

Reviewer 1 Report

Seems like an interesting research question. Well-written paper.

There are some scattered typos, such as missing dots.

ABSTRACT

The Abstract needs quite a bit of work. There is no equivalent of Introduction, telling us why this is an interesting topic, and giving us the theoretical background. There are even no Research Questions.

We immediately head into the equivalent of Method.

The equivalent of Results focuses entirely on significance testing. With the huge sample sizes, even uninteresting results become significant. It is the equivalent of losing half a gram of fat more in a comparison of two different diets, both taken by a sample of one million. The authors should carefully read the influential paper by Funder & Ozer (2019) and base their conclusions upon effect sizes, such as r or d.

The equivalent of Conclusions comes out of the blue, because the theoretical underpinning is not given at the beginning of Abstract.

INTRODUCTION

A useful review of the literature.

  1. 39 ‘extreme measures’; extreme in comparison to what? Please argue why these measures are extreme. Isn’t it good to have strong measures in a dangerous situation? The word ‘extreme’ has a negative connotation. Maybe choose another word?
  2. 51-55. A Hegelian sentence. Break up into two sentences for legibility. [minor point]
  3. 58 ‘rich debate’; can a debate be rich?
  4. 79 Different letter type – makes the text look messy.

Would the authors say that this study constitutes a so-called ‘natural experiment’? If yes, that would have implications for the title of the study.

  1. 98-101. Rewrite and replace significance levels by effect sizes, based upon the existing effect sizes in the literature. Significance depends upon sample size.

METHOD

Great sample.

  1. 105 what is a ‘paired sample’? Please explain.
  2. 119 date were …. Data = plural

Good reliabilities of the instruments.

RESULTS

The data were competently analyzed.

DISCUSSION

In Discussion authors should tell to what degree their hypotheses were conformed, using non-statistical language. The beginning of Discussion is too close to Results. Reframe the beginning of Conclusion in effect sizes. See reference below.

Interesting Discussion. Feel free to expand. Can findings be generalized to other countries with different populations and different cultures? There are many different groups in Israel – do the findings apply to all groups?

ENCOURAGEMENT. The paper needs a bit of brushing up, but fundamentally it is sound. Good luck with the revision.

LITERATURE

Funder, D. C., & Ozer, D. J. (2019). Evaluating effect sizes in psychological research: Sense and nonsense. Advances in Methods and Practices in Psychological Science, 2, 156-168.

Reviewer 2 Report

COVID-19 has impacted all of us, not only in terms of physical health but also overall health and standing. The authors study distress, resilience and subjective well-being of the Jewish Israeli population using data from two different time periods after CVOID-19 and compare it to prior data. The manuscript is well written and is easy to follow. As someone who is familiar with the supply chain resilience, I find it interesting to read about resilience at different levels. Additionally, the situation Israel had with the elections puts a unique position for the impact of COVID-19 on Israel. I thank the authors for this interesting perspective and the paper.

My overall questions is about the framework how the three groups of indicators are studied. These factors, at least the subjective well-being and even the distress response (as a result) depend on the underlying health conditions of the responder. It is known that there are high risk populations such as populations with underlying health conditions and immunocompromised people for COVID-19. It is logical to expect these populations to have lesser scores for subjective well-being and have more anxiety towards COVID-19. Was this data collected and/or was the prior health of the survey population a good representation of the general population? If not, was the reason why the authors only hypothesize correlation rather than causation Depending on the data, I feel like controlling for prior health (or having a different approach as thinking as health as a mediator or moderator) might be considered for better representation of the relationship among these factors.  

What happened to the 18% who were not included in the second round? What are the reasons for not being involved? Being sick/hospitalized during the second wave of the interviews? Would there be any deaths?

Formatting of Line 79-83 needs attention.

When you are explaining study tools, especially the order of the variables used, it might be better to use the order in Table 2 for consistency.

Reviewer 3 Report

The idea of the paper is very interesting and a proper proposal in these days. I have found several problems and I would like to make a few comments about the article:

Abstract

The abstract starts with methods, without information abut the background of the study (lines 15-16)

There is no evidence in the study for the abstract´s conclusions (line 26-27)

It seems to be necessary a connection in the text between lines 41-42.

Introduction

Authors use Subjective wellbeing as a synonim of Perceived wellbeing. I am not sure about it, becuase perceived wellbeing can be a interjudge concept but subjective wellbeing is an individual perception. I suggest to use one in the text (better subjective, that is defined in the Introduction).

Methods

What are the aims of the study? There is no specific part for them in the text

It´s necessary more information about the design of the study (lines 104-105). Directly it starts with the data collection, with no longer explanation than "longitudinal study".

Also it seem sto be necessary a brief explanation about data analysis, how has been done and what kind of statistics hav been used for it.

Results

Best part of the paper. Really interesting.

Discussion.

It´s absolutely necessary a part of biases and limitations of the study, especially in this study with a lot of references to cultural topics and biases.

Conclusions.

There is no connection between lines 342-345 conclusion and the data of the study. It´s possible and logical, but it´s not connected with data and results.

Round 2

Reviewer 1 Report

The authors did a good job processing my feedback.

Reviewer 3 Report

I would like to appreciate the work of the authors in the second verssion of the paper. Concrete, proper and very clear. Congratulations